# The Sub-Molecular and Atomic Theory of Cancer Beginning: The Role of Mitochondria

**DOI:** 10.3390/diagnostics12112726

**Published:** 2022-11-08

**Authors:** Mario G. Balzanelli, Pietro Distratis, Rita Lazzaro, Van H. Pham, Toai C. Tran, Gianna Dipalma, Francesco Inchingolo, Diego Tomassone, Sergey K. Aityan, Sossio Vergara, Kieu C. D. Nguyen, Ciro Gargiulo Isacco

**Affiliations:** 1SET-118, Department of Pre-hospital and Emergency, SG Giuseppe Moscati Hospital, 74120 Taranto, Italy; 2Nam Khoa Biotek, Ho Chi Minh City 70000, Vietnam; 3Department of Histology-Embryology and Genetics, Pham Ngoc Thach Medical University, Ho Chi Minh City 70000, Vietnam; 4Department of Interdisciplinary Medicine, School of Medicine, University of Bari “Aldo Moro”, 70124 Bari, Italy; 5Foundation of Physics Research Center, 87053 Celico, Italy; 6Multidisciplinary Research Center, Lincoln University, Oakland, CA 94612, USA; 7LCS Mameli, 00118 Roma, Italy

**Keywords:** atoms, protons, neutrons, electrons, mitochondria, cancer

## Abstract

Life as we know it is made of strict interaction of atom, metabolism, and genetics, made around the chemistry of the most common elements of the universe: hydrogen, oxygen, nitrogen, sulfur, phosphorus, and carbon. The interaction of atomic, metabolic, and genetic cycles results in the organization and de-organization of chemical information of what we consider living entities, including cancer cells. In order to approach the problem of the origin of cancer, it is therefore reasonable to start from the assumption that the atomic structure, metabolism, and genetics of cancer cells share a common frame with prokaryotic mitochondria, embedded in conditions favorable for the onset of both. Despite years of research, cancer in its general acceptation remains enigmatic. Despite the increasing efforts to investigate the complexity of tumorigenesis, complementing the research on genetic and biochemical changes, researchers face insurmountable limitations due to the huge presence of variabilities in cancer and metastatic behavior. The atomic level of all biological activities it seems confirmed the electron behavior, especially within the mitochondria. The electron spin may be considered a key factor in basic biological processes defining the structure, reactivity, spectroscopic, and magnetic properties of a molecule. The use of magnetic fields (MF) has allowed a better understanding of the grade of influence on different biological systems, clarifying the multiple effects on electron behavior and consequently on cellular changes. Scientific advances focused on the mechanics of the cytoskeleton and the cellular microenvironment through mechanical properties of the cell nucleus and its connection to the cytoskeleton play a major role in cancer metastasis and progression. Here, we present a hypothesis regarding the changes that take place at the atomic and metabolic levels within the human mitochondria and the modifications that probably drive it in becoming cancer cell. We propose how atomic and metabolic changes in structure and composition could be considered the unintelligible reason of many cancers’ invulnerability, as it can modulate nuclear mechanics and promote metastatic processes. Improved insights into this interplay between this sub-molecular organized dynamic structure, nuclear mechanics, and metastatic progression may have powerful implications in cancer diagnostics and therapy disclosing innovation in targets of cancer cell invasion.

## 1. Introduction

### 1.1. The Connection between Cell Nucleus and Mitochondria: The Possible Origin of Cancer

It is widely accepted that mitochondria likely evolved from engulfed prokaryotes that once lived as independent organisms forming an endosymbiotic relationship and gradually developing into a mitochondrion as we know today. Mitochondria divide independently from the cells adopting a similar prokaryote mechanism. Specifically, mitochondria reproduce within the cell to be relocated when cell division is completed. The transition from endosymbiotic bacterium to permanent organelle went through a long sequela of evolutionary changes that involved the generation of hundreds of new genes and new amino acid and protein production systems, genes transferring, lateral gene transfer insertion of membrane transporters, genome reduction, and the retargeting of proteins [1,2].

Interestingly, several eukaryotic lineages have adapted to survive in low-oxygen conditions, either in water or terrestrially, or even in animal gastrointestinal tracts. Within poor aerobic respiration, many of these organisms have evolved mitochondria with reduced, or no, cristae that function anaerobically. Depending on the grade of these adaptations, some scientists grouped the mitochondria into five types based on their energy metabolism: aerobic, anaerobic, hydrogen-producing, hydrogenosomes, and mitosomes [3]. 

The outer and inner membrane play a key role in mitochondrial homeostasis. The outer membrane covers the mitochondrial matrix and is furnished with pores that allow the free passage of ion proteins; the inner membrane works as a cell membrane, involved in electron conveyance and ATP synthesis (oxidative phosphorylation). Within the inner membrane takes place the production of electrons in which oxygen is conveyed in forming water, H_2_O. During this process, the involved proteins drive protons out of the matrix into the intermembrane space to generate an electrochemical gradient. The oxygen is extremely important for ATP synthesis and chemiosmosis. The lack of ATP makes cells stop functioning and die [4,5,6].

Mitochondrial damages due to oxidative stress and changes in ion position have been implicated in several diseases as cancers and neurodegenerative diseases, including Alzheimer’s disease (AD), Parkinson’s disease (PD), and amyotrophic lateral sclerosis (ALS). Until now the manifestation of oxidative stress characterized by the overproduction of reactive oxygen species (ROS) was considered the main cause of mitochondrial DNA mutations involving mitochondrial respiratory chain damages, membrane anomalous permeability, and influence Ca^2+^ homeostasis and mitochondrial defense systems [1,2,3,4,5,6].

### 1.2. The Cancer Cell Structure Seen from Physics Perspective

An average of 10^14^ atoms are present in a typical human cell and the number of cells in the human body is probable the same number. Perhaps the best way to find a successful tool to fight cancer is to understand its basic atomic composition, starting from the foundation of visible world, “matter”. Matter is a complex system made of almost infinite combinations of elements—substances such as hydrogen or carbon, ultimate pieces of life that cannot be further fragmented by any chemical means. The atom constitutes the smallest particle of any element that still retains a distinctive chemical property. Traits, features, and specific characteristics of molecules, including the materials from which living cells are made, would eventually rely on a highly structured combination of atoms. Therefore, highlighting the inanimate deep core of living organisms understanding the inner chemical bonds that keep together or shred atoms apart, it is crucial in oncology science [7,8,9].

A positively charged nucleus Is the core of each atom, the nucleus is composed by protons, positively charged (which indicate the atomic number) and neutrons, electrically neutral. Each nucleus has satellites negatively charged known as electrons held in circular orbital tracks by electrostatic attraction force. Hydrogen atoms, for instance, have a nucleus composed of a single proton, which indicates the atomic number of 1, and is the lightest element, whilst the carbon atom nucleus is composed of six protons which indicates an atomic number of 6. The numbers of negatively charged electrons are always equivalent to the number of positively charged protons within the nucleus. The electrons indicate not only the atomic number but determine the chemical behavior of an atom as well [7,8,9].

Molecules are made of atoms; thus, it is almost inevitable not to focus on the reciprocal influence between the atomic structure and the bio-chemical mechanisms seen as new scientific trend considering the correlation between systems, organs, cells, cell components/molecules, and atoms. This new branch of medicine should be based on what we may call the “atomic medical-biology” following Tofani Santi’s indications. Tofani Santi coined physical biology with the intent of clarifying the way atomic events affect biochemical reactions and cell life, and the genesis of degenerative diseases such as cancer. In cancer, medicine, chemical biology, and molecular biology have been unable to fully satisfy therapeutic needs [7]. 

For example, the use of magnetic fields revealed the direct influence on atomic energy levels, a support that gives to medical doctors the use of magnetic resonance imaging (MRI) considered a powerful tool in understanding molecular/atomic dynamic environment of any biological structure, reaching a precision and accuracy never experienced before. Chemistry explained how electron spin state has a pivotal role in all the reduction-oxidation reactions essential in cellular metabolic pathway, governing the behavior of the biological system, influencing genetic stability. The synthesis of many complex molecules often requires the oxidation of their precursor, via the use of molecular oxygen, the ready-to go electrons are needed in reassigning procedures when electron spins assume specific energy levels. Quantum physics helps to clarify these energy levels, explaining how magnetic fields may eventually interfere within these processes. The point of view of physics may allow the use of specific static magnetic fields in similar way to gravity. In fact, the electromagnetic field tensor FμνFμν, which explains all the information about both electric and magnetic field, surely contributes to the energy-stress tensor *TμνTμν*, which appears in the Einstein Field Equations:*Gμν = 8πGTμνGμν = 8πGTμν*

This equation encodes the geometry of space-time, while it describes the “sources” of gravity. Moreover, from gravitational redshift/blueshift and the law of conservation of energy, the obtained equations are:Δg = fG/μ0B and Δg = fGε0E

These equations confirm the close interaction between gravitational and magnetic/electric field [7,8,9,10].

In fact, each cell is composed of an almost infinite number of atoms that have at their center a positively charged nucleus surrounded by a ring of negatively charged electrons, held in a series of orbitals by electrostatic attraction to the nucleus. It is the balance that follows nucleus’s protons and neutrons equilibrium that determines whether a nucleus will be stable or unstable. An atom becomes unstable or radioactive if the nucleus has an excess of internal energy that results from an excess of either neutrons or protons. We assume that the combined gravitational force from the protons and neutrons in a nucleus implies the existence of an additional attractive force similar in size to the electrostatic repulsion, which eventually holds the nucleus and mitochondria together and keeps the stability of cells. Normal cell activity requires timely and accurate “in-out” transmission of information from receptors located alongside cell membrane (CM) to the nucleus and mitochondria [8,9].

Gatenby and Frieden proposed the intracellular electric field as core of the information transfer, which is generated by distribution of charge on the nuclear membrane. The electrostatic force between point charges *q1* and separated *q2* (the net electric charges of the two objects) by a distance *r12* (the vector displacement from *q1* to *q2*) is given by Coulomb’s law. Newton’s third law that states “every force exerted creates an equal and opposite force” applies as usual—the force on *q1* is equal in magnitude and opposite in direction to the force it exerts on *q2* (Figure 1) [8,9,10,11]. 

On average, a billion cells inside the human body replicate every day, which includes almost a thousand billion DNA bases replication during each cell division. This check-point function is extremely well organized by blocking and repairing errors and transcription mistakes every second. On the other hand, the accumulation of minor errors may lead to a serious effect on DNA integrity [9]. The connection bridge between DNA integrity and cellular atomic integrity may reside on the principle of ionization energy (IE), considered the amount of energy needed for a gaseous atom in ground electronic state to discharge an electron (Figure 2). When ionizing takes place, it may affect the cells and several things can happen that damage the cells by preventing DNA from replicating and folder properly. During this process, the atom that has lost an electron becomes positively charged (positive ion) while the atom that has acquired an extra electron becomes a negative charged. Both negative and positive ions are extremely reactive by interacting with all the molecules around them. The atom that has lost or acquired an electron induces the molecule to become either positively or negatively charged. Such molecules are known as free radicals; due to their charge and their reactivity, these molecules attract electrons from nearby atoms. This mechanism gives rise to new molecules, often unstable and triggering a chain reaction that can deeply damage cell structures. Among the best-known free radicals, there are those that contain at least one so-called oxygen atom, ROS (Reacting Oxygen Species). The generation of free radicals occurs physiologically in the nuclear reactions, either due to environmental or endogenous factors. In addition, a molecule that has kept an atom together with one inside of it of different charge can cause the change of the shape of the final protein with a different chemical behavior [10,11].

This process may eventually generate dysfunctions during different processes such as phosphorylation, methylation, hydrolysis, and oxidation which furtherly affect DNA. This damage may occur any time at any significant level in vivo due to reactive metabolites continuously being generated at different physiological stages and partly in generating abnormal proteins. Damages in the DNA may induce a chain of reaction that blocks important cellular processes such as protein replication and transcription gene sequences and gene expression leading to cancer, neurodegenerative disorders, and biological aging [10]. 

Though cells use different biological strategies and tools to overcome DNA damages, repairing mechanism, apoptosis, and autophagy may only work for a limited time and with only a limited number of errors. Apoptosis appears to be one of the most interesting and the major processes involved in the stability of matter determined by the electron spin at least within the cell nucleus. By adopting apoptosis, cells may control the genetic machinery to avoid an altered DNA replication [11,12,13,14]. 

The main question at this point is, where do the cancer cells come from and what are they really? It is widely accepted that cancer is a non-homogenous disease that grows and develosp linearly, composed of sequential phases due to several causes. Tumorigenesis consists of many distinct patterns and is mainly a complex process influenced by a multitude of factors that are linked to genetic predisposition, age, immune system, nutrition, and both macro and micro-environmental influences. Cancer can only be defeated by recognizing the basic heterogeneity of its molecular and atomic structure, highlighting the aberrations that make cancer cells almost invincible once compared to normal cells [7,8,9,10,11].

A crucial contribution in trying to explain this assumption came from Albert Szent-Györgyi. There was the necessity to cross over the molecular biology’s soluble proteins principle and start exploring cell activity on a submolecular level by understanding the electron transfers between molecules to seek to explain what causes the normal cells to deviate from normal proliferating [12]. Based on this stance, we should refer to a “primordial state”, in which primordial microorganisms’ primary functions were based on fermentation to live and proliferate. Oxygen was the turning point that changed everything inducing proteins’ new behavior to link together into increasingly complex systems reaching what the author called “beta state” [12]. However, the proliferation mechanism changed as well, the cells at this stage had to switch back, from methylglyoxal pathway via glyoxylase (an enzyme present in all living cells) to the primordial state and start dividing. Intriguingly, in normal conditions, this enzyme system is self-reversing; nonetheless, the shortage of methylglyoxal or an excess of glyoxylase might keep the cell in a permanent proliferative state, similarly to what takes place during the cancer proliferative mechanism [12]. Starting in 2000, the interest in searching for some physical or mechanical difference that could help distinguish the two types of cells, cancerous and normal cells, researchers found that the surface coat surrounding cancer cells revealed markedly different features, later referring to “fractal dimensionality.” Fractals occur often in nature as consequence of chaotic behavior and cancer has been associated with chaos as well. Those effects, as proposed by Basov et al., were observed mainly at the mitochondria and cell levels as a new hypothesis for heavy nonradioactive isotope fractionation in living systems via neutron effect realization [13].

Human cervical epithelial cells were detected by a high-resolution AFM procedure, The largest changes were observed at the scale of surface features ranging between 1 and 300 nm. There were clusters of molecules and microvilli/microridges and the changes in fractal geometry on the cell surface during their progression to cancer was assumed to refer to these aggregates or cluster [14,15,16,17,18]. 

Elkington et al. compared the variation of different stages of cancer versus normal tissues from the same organ: pancreatic, breast, colon, and prostate. The overall results confirmed the fractal evolution of cancer from stage I with an increase of 2%, stage II with an increase of 4%, and stage III of 7% compared to the normal (*p*-values < 0.05) (Figure 3) [18]. 

Though morphological changes are seen in different cancer cells, it is interesting to observe that these cells show a negative cell surface charge metabolically regulated by glycolysis, a dynamic process depending on the level of glucose provided. The negative charge generated on the cancer cells due to a specific sugar metabolism pathway and morphology changes are shown by the very atypical pericellular brush distinctive of normal cells turning into cancerous [15,16,17,18,19]. 

## 2. Cancer Seen as a Mitochondrial Survival Way-Out

There are many studies on the cancer surface biophysical behaviors regarding fundamental oncology and clinical diagnosis and treatment. Functional ATP in normal tissues is obtained from oxidative phosphorylation while the remaining 10% c.ca is from aerobic glycolysis known as the Pasteur effect. Cancer cell metabolism and ATP are greatly based on glucose usages even under adequate supply, otherwise known as the Warburg effect. Therefore, cancer cells’ demand for glucose is almost unquenchable and the 18F-fluorodeoxyglucose contrast agent (FDG) is developed to track the position of glucose in the human body, detected by positron emission computed tomography (PET) scanner (Figure 3) [20]. 

Altered energy metabolism consisting of increased resting energy expenditure associated with an augmented metabolism of sugar, lipid, and proteins are changes viewed as cancer-typical alterations in intermediary metabolism. Cancer cells are able to ferment glucose in the presence of oxygen, suggesting defects in mitochondrial respiration mechanism as potential cause of cancerous metamorphosis. Cancer cells both in vivo and in vitro, secrete large amounts of lactate ions as mobile anions up to 30 times higher than in normal cells, ions that can cross the plasma membrane inevitably change the surface charge. Cancer cells work in complete autonomy promoting biosynthesis and cell growth using the residual organic carbon to generate cellular biomolecules. In addition, glycolysis allows tumor cells to overcome normal cells or immune system cells for glucose uptake, and the increase of lactic acid causes changes in the tumor cells’ micro-environment that enhance tumor cell growth and spread. Therefore, anything that disrupted the charge-transfer processes in cells structural proteins might push cells into the “primordial state” of continuing proliferative mode [12,20,21].

Cancer as a disease is strictly associated with the changes of the modern industrialized world. Modifications in nutrition, environmental changes, and stress actively contribute to the course and rate of cancer diseases. In normal conditions, free radicals (atoms or molecules with unpaired electrons) generated by oxidation-reduction reactions accumulated within cells are well controlled and inactivated by specific enzymes such as mitochondrial carbonic anhydrases [22,23]. However, free radicals are also accumulated via environmental pollutants such as food, water, smog, smokes, drugs, chemicals, or radiation. Free radical structure is composed of unpaired electrons which makes them extremely reactive, inducing them to steal electrons from other molecules, leading to bond breakages including the covalent bonds in enzymes and other proteins, DNA, and the lipids in cell membranes procuring unrepairable errors. Thus, modern metabolic disorders and degenerative diseases such as diabetes, neuro-degenerative disorders, and cancers are characterized by common pathological features linked to the increase of intracellular oxidative stress which in turn leads to earlier cell apoptosis and abnormal limitless proliferation. Mitochondrial carbonic anhydrases have been shown to be important mediators of tumor cell pH by modulating the bicarbonate and proton concentrations for cell survival and proliferation [23,24,25].

Therefore, we extended our current analysis to determine the effect of these effectors on the atomic core of mitochondria as consequence of long-term increase in cell respiration, carbonic anhydrase, and ROS accumulation. We propose that those factors would play a significant role in eliciting functional changes in the magnitude of electric field at the mitochondria atomic level by increasing the accumulation of electrons, following Coulomb’s law (Figure 4). 

However, why mitochondria? For mitochondria in particular, data suggested that these organelles could carry and transfer wicked information, leading to metastases avoiding the immune system checkpoints and apoptosis. The nucleus controls the proteins and information transmitted to the mitochondria by anterograde regulation that reflects different stressors through the nuclear genome reprograming which eventually modulates mitochondria biogenesis. Identifying whether specific metabolic alterations drive or merely follow the atomic devolution of cancer cells is not a small question, as in both cases, targeting these changes may block phenotypic progression and metabolic plasticity [26,27,28]. Mitochondria need both mitochondrial and nuclear gene products; however, the interesting part is the way these organelles replicate by dividing in two, similarly to cell division employed by bacteria, mitochondria originate only from other mitochondria [26,27,28,29]. 

Mitochondria contain their own DNA, circular as the bacteria have, along with their own transcriptional, translational machinery, and ribosomes. The transfer of RNA molecules takes place in a similar process to those of bacteria, as they are components of their membrane. The endosymbiotic hypothesis suggests that mitochondria are originally from specialized bacteria that somehow successfully homed into the cytoplasm of a different species of prokaryote or some other cell type commencing a sort of reciprocal favorable long-term coexistence. 

The evolutionary success was based on the ability to conduct cellular respiration in the host cells by glycolysis and fermentation; the form in which the bacteria and cells could both survive the time would have been enormously based on evolution [29]. 

Yet, as the presence of mitochondria in the eukaryote common ancestor kept changing until present day as the integral part of human cell biology, they also definitively reflect their ubiquitous role in the life of host cells preserving a kind of original independent status. Therefore, we proposed the hypothesis that cancer cells could be mitochondrial mutation cells as the way to survive a hostile microenvironment, the dysfunction of IE as consequence of age, environmental insults and metabolic deviances (free radicals-ROS, smoke, pollutants, food, UV radiation, etc.). Furthermore, the accumulation and the constant increase of ROS inevitably leads to abnormal coupling processes that dissociate the electron transport chain from phosphorylation by ATP-synthase, preventing the formation of ATP and switching into sugar anaerobic energy mode which is typical of cancer cells’ survival system. The damage of the phospholipid bilayer of mitochondrial membranes leads to an atomic disorganized state, which allows protons to flow through without control weakening the electrochemical gradient and transferring protons without the use of ATP-synthase such that no ATP is produced [29,30,31]. 

The mitochondria’s ability to completely detach from the host cells has to be seen as an integral part of mitochondria DNA original code as mitochondria are able to undergo structural and functional remodeling via the transmission of signals to downstream executioner proteins and profound isotopic exchange. This pathway may take place through death stimuli such as dioxygen, metabolic perturbation, deprivation of survival factors, oxidative stress, Ca^2+^ overload, DNA damage, proteotoxic stress, and oncogene activation. While the cell becomes starved of ATP, we propose that mitochondria slowly regress to their “original primordial” as previously supposed by Albert Szent-Györgyi, starting to adapt to use fermentation as if in anaerobic conditions; this may cause a type B lactic acidosis in affected patients [30,31,32]. 

Particularly, isotope exchange reactions in eukaryotic systems can be accompanied by deep structural changes that affect the rate of biochemical reactions at the molecular level. This is clear at the level of energy metabolism mechanism in mitochondria as Basov et al. described [13,33,34]. For instance, the replacement of deuterium by protium follows the acceleration of proton fluxes within mitochondria driving to higher resistance of the cell to events such as hypoxia and intoxication. Results were confirmed by recent outcomes in which the deuterium was seen as an important effector of metabolic activity in stem cells’ differentiation process with a decrease in the effectiveness of adipogenic differentiation, probably associated with mitochondrial dysfunction [33,34]. 

## 3. Conclusions

The main intent of this article was to help refocus the atomic importance and role of mitochondria in the general pathophysiology of cancer. We have therefore attempted to suggest that cancer cells are the final results of a long-term effort to face internal and external stimuli that progressively destabilize the integrity of the cells. We proposed that cancer cells are of mitochondrial origin, a sort of mutation/solution that mitochondria adopt to survive a general cellular decay which explain their absolute capacity to respond to all immunity attacks and to adapt to all adverse circumstances. From our perspective, a significant point was to determine and characterize whether atomic, metabolic, and genetic alterations could possibly be categorized in the whole process of malignant development. The literature seems to support this theoretical line in which human mitochondria are able to use adaptive survival mechanisms very similar to bacteria. If this were confirmed, this would redirect the therapeutic strategy towards new and more appropriate approaches to treat cancers.

## Figures and Tables

**Figure 1 diagnostics-12-02726-f001:**
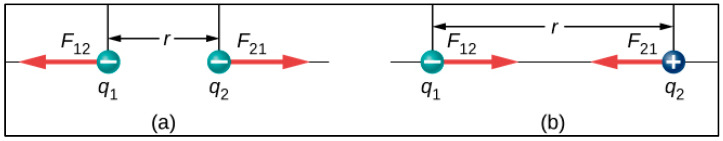
k = 8.99 × 109 N·m^2^C^2^, a universal constant called the Coulomb constant, q1 is the charge of particle 1, q2 is the charge of particle 2, and r is the distance between the two particles. (**a**) Like charges; (**b**) unlike charges.

**Figure 2 diagnostics-12-02726-f002:**
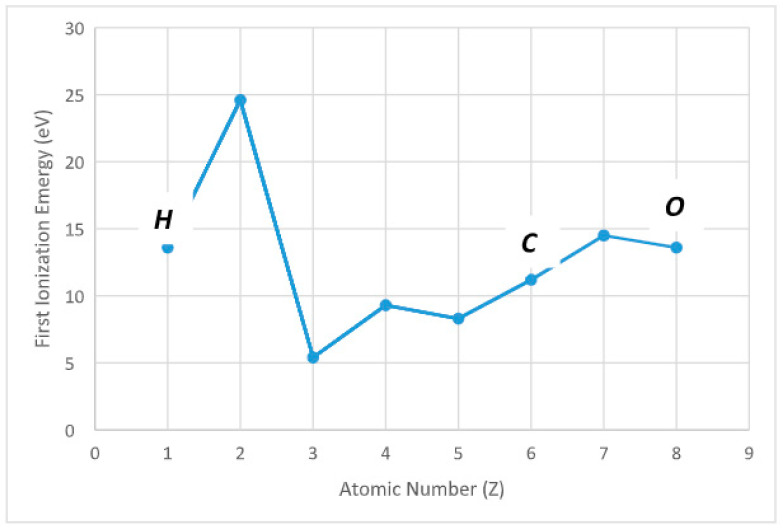
Ionization energy for main elements (hydrogen, carbon, oxygen) in cell chemistry. Ionization energy is the amount of energy that an isolated gaseous atom in the ground electronic state must absorb to discharge an electron, resulting in a cation (x = atomic number) (y = IE quantity needed by the atom to detach the electron).

**Figure 3 diagnostics-12-02726-f003:**
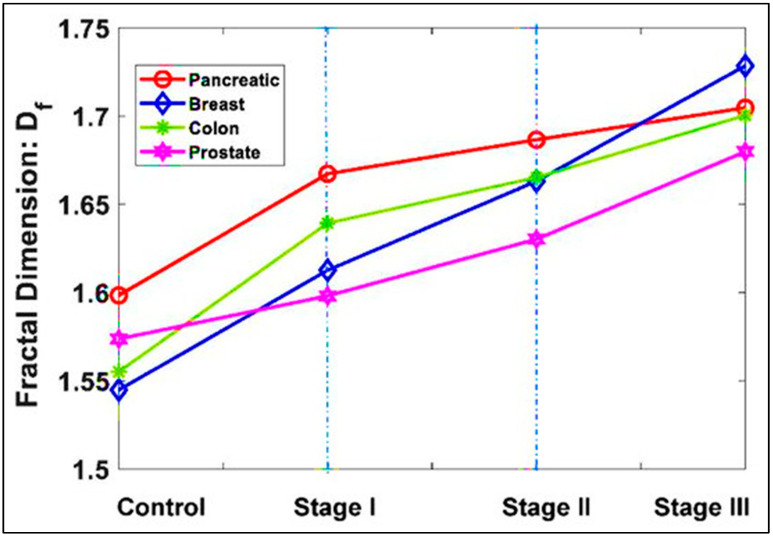
The mean values of the fractal dimension of different cancer for different stages. Normal Stage I Stage II Stage III Pancreatic 1.5984 1.6673 1.6866 1.7047 Breast 1.5448 1.6126 1.6631 1.7283 Colon 1.5551 1.6393 1.6652 1.7004 Prostate 1.5737 1.5981 1.6302 1.6798. Permission has been obtained from the article Fractal Dimension Analysis to Detect the Progress of Cancer Using Transmission Optical Microscopy Authors [18].

**Figure 4 diagnostics-12-02726-f004:**
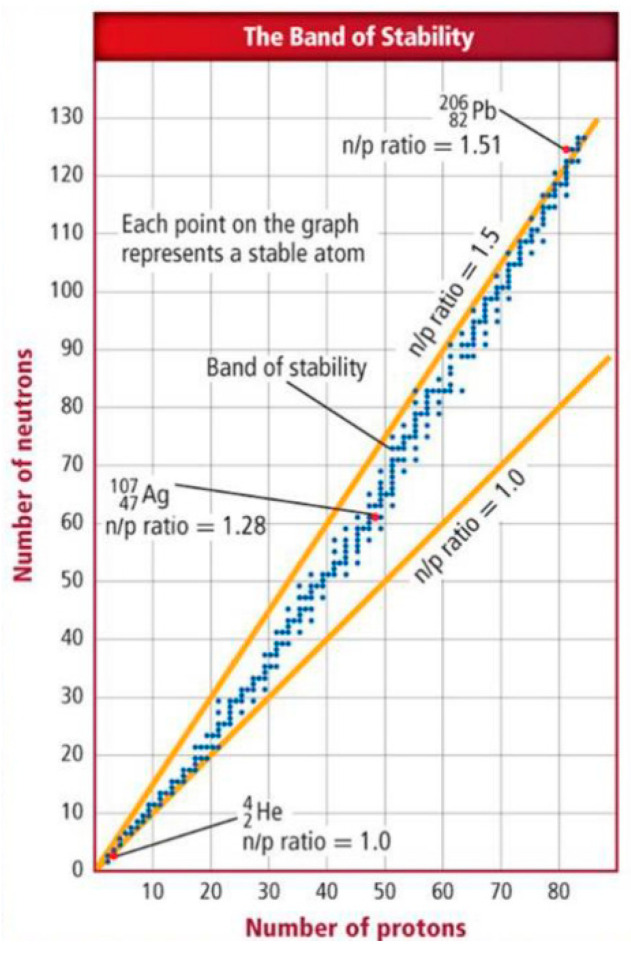
The graph shows plot of the number of neutrons versus the number of protons in various stable isotopes. The stable nuclei (pink band) have a neutron/proton ratio between 1:1 and 1.5. As the nucleus gets bigger, the electrostatic repulsions between the protons gets weaker. The nuclear strong force is about 100 times as strong as the electrostatic repulsions. It operates over only short distances. After a certain size, the strong force is not able to hold the nucleus together. Adding extra neutrons increases the space between the protons. This decreases their repulsions but, if there are too many neutrons, the nucleus is again out of balance and decays (wps.prenhall.com). Accessed on 24 September 2022.

## Data Availability

Not applicable.

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
