# Peer review of "The Sub-Molecular and Atomic Theory of Cancer Beginning: The Role of Mitochondria"

_diagnostics, 2022, doi:10.3390/diagnostics12112726_

Round 1

Reviewer 1 Report (Previous Reviewer 2)

With current modifications, it sounds better.

Reviewer 2 Report (Previous Reviewer 1)

Great job.

This manuscript is a resubmission of an earlier submission. The following is a list of the peer review reports and author responses from that submission.

Round 1

Reviewer 1 Report

General comments to the paper entitled

The Sub-Molecular and Atomic Theory of Cancer Beginning: the Role of Mitochondria

I was pleased reading the title of the paper but a little bit disappointed with the content of it.

Talking about sub-molecular processes the authors first should mention Albert Szent-Györgyi, who first suggested “there is another dimension below the molecules”.

(“Our body is built of molecules, so its reactions have to be molecular reactions, but the molecules are built of atoms, and atoms are built of nuclei and electrons. So, there is another dimension below the molecules which has been disregarded by biology.” International Journal of Quantum Chemistry, 7, 217-223 (1980); Quantum Biology Symposium)

It was over 40 years ago and at that time we considered „theory”, but today after the progress in molecular biology, gene mapping and sequencing, discovering the critical signal transaction pathways, identifying thousands of gene alterations in cancer cells, the role of ROS in cell metabolism it is an expectation of the paper addressing to sub-molecular processes to cover the key results science has achieved in the last 40 years.

The paper does not contain any new information or even does not mention anything about the recently published papers about the sub-molecular mechanisms which are related to electron transport and deuterium/hydrogen ratio.

Author Response

Answer to the Reviewer

We really appreciated the comments of the reviewer. We added into our article the precious information as suggested by the Reviewer of Albert Szent-Györgyi, who first suggested “there is another dimension below the molecules”, in addition we added some more information that are clearly visible in the references and main text highlighted in yellow (12,13,30-34). Said that, we would like to make our  point of view clearer which is the follow: the innovation approach of this Hypothesis article is something that link together the Atomic instability theory *isotope shock* and the “independent nature” of mitochondria as main causes of cancer genesis, its proliferation and progression within the host. Certainly we are well aware that this is just an assumption and remain at the moment just a theoretical assumption...However, the consequence of putting together the three planes, biology, physics and microbiology as well as the correlation between primordial bacteria, mitochondria and cancer  still remain something very little investigated and new.  We do believe that Cancer Cells are strictly related to Mitochondria, indeed we do believe that Cancer Cells are Mitochondria that escaped apoptosis and survived after having reprogrammed their own DNA.

Reviewer 2 Report

The paper itself is ok, however, it would be beneficial for the conclusion and for the completion of the paper to evaluate in parallel with the presence of genetic/protein alteration in mitochondria.

Author Response

Answer to the Reviewer

We thank you for your valuable suggestion, some corrections have been made but we have chosen not to include the part on mitochondrial genetic alterations, in order to reserve this important in-depth study for other, more detailed papers.
